# Correlation holes and slow dynamics induced by fractional statistics in gapped quantum spin liquids

Oliver Hart [1], Yuan Wan[2,3] & Claudio Castelnovo [1✉]

Realistic model Hamiltonians for quantum spin liquids frequently exhibit a large separation of energy scales between their elementary excitations. At intermediate, experimentally relevant temperatures, some excitations are sparse and hop coherently, whereas others are thermally incoherent and dense. Here, we study the interplay of two such species of quasiparticle, dubbed spinons and visons, which are subject to nontrivial mutual statistics – one of the hallmarks of quantum spin liquid behaviour. Our results for $\mathbb{Z}_2$ quantum spin liquids show an intriguing feedback mechanism, akin to the Nagaoka effect, whereby spinons become localised on temperature-dependent patches of expelled visons. This phenomenon has important consequences for the thermodynamic and transport properties of the system, as well as for its response to quenches in temperature. We argue that these effects can be measured in experiments and may provide viable avenues for obtaining signatures of quantum spin liquid behaviour.

[1] T.C.M. Group, Cavendish Laboratory, Cambridge, United Kingdom. [2] Institute of Physics, Chinese Academy of Sciences, Beijing, China. [3] Songshan Lake Materials Laboratory, Dongguan, Guangdong, China. ✉email: cc726@cam.ac.uk

opologically ordered phases of matter have attracted much attention over the past few decades[1,2] thanks to their unusual behaviour, which is of fundamental interest and has potential applications in quantum information storage and processing[1,3,4]. Such states are characterised for example by subleading corrections to the ground state entanglement entropy[5,6], and by a ground state degeneracy that depends on the genus of the space on which the system resides[7]. Their low-energy excitations often take the form of pointlike, fractionalised quasiparticles with anyonic statistics[8].

While concrete and unambiguous experimental evidence for these unusual ground state properties remains in general unavailable, the exchange statistics of the quasiparticles and their fractional quantum numbers offer some of the most promising routes to unique and experimentally accessible signatures of topological order[9,10]. Examples of such excitations include the Laughlin quasiparticles of the fractional quantum Hall effect[11] or the Majorana fermions in Kitaev-like materials[12].

In the context of quantum spin liquids (QSLs)—topologically ordered phases that arise in frustrated magnets at low temperatures[13–15]—we reflect on the fact that realistic model Hamiltonians exhibiting QSL behaviour can often be constructed with[16,17]: (i) a large, classical constraint that projects the Hilbert space onto an extensive set of local tensor product states; and (ii) quantum fluctuations. The fluctuations induce coherent superpositions of the tensor product states, endowing the system with quantum topological properties, but must not be strong enough to drive the system across a confinement/Higgs transition. In such systems, there are quasiparticles (that we dub spinons) that violate the classical constraint; these have a large energy cost $\Delta_s$ and smaller but significant hopping matrix elements of magnitude $t_s < \Delta_s$ (typically of the order of the quantum fluctuation—e.g., exchange—terms present in the system). There are also gapped excitations, which we dub visons, that disturb the quantum phase coherence amongst the constrained states, whose energy cost $\Delta_v$ is perturbative in $t_s/\Delta_s$ in the deconfined phase and thence much smaller than both $\Delta_s$ and $t_s$. Typically, the characteristic magnitude of their hopping matrix elements is smaller still, $t_v < \Delta_v$. A case in point is indeed quantum spin ice[14] with small transverse terms. While this may not be considered an example of topological quantum order per se, its microscopic Hamiltonian is nonetheless an example of how one could realise a QSL in experiment. It features a large projective energy scale and small transverse kinetic terms, which give rise to an eminently accessible temperature range where the results in our paper apply.

In this scenario, it is of experimental interest to consider the temperature range where

$$t_v < \Delta_v \lesssim T \ll t_s < \Delta_s. \qquad (1)$$

Upon cooling the system, it is the highest temperature at which one can hope to observe signatures of QSL behaviour. Any precursor diagnostics in this temperature regime would be greatly beneficial before attempting to reach challengingly low temperatures where both quasiparticle species behave quantum coherently ($T < t_v$). In the temperature range given by Eq. (1), visons are thermally populated with a finite density, whereas spinons are sparse and hop coherently across the system on a timescale $O(1/t_s)$ that is fast with respect to the stochastic motion of visons, which occurs on a timescale $O(1/t_v)$ or longer. It is then natural to take a Born–Oppenheimer perspective and treat the visons as static quasiparticles when considering the motion and equilibration of spinons. The slow dynamics of visons allows parallels to be drawn with Falicov–Kimball models[18,19], and models of quasi-MBL[20–22] and disorder-free localisation[23].

We focus on the case of a $\mathbb{Z}_2$ topological spin liquid, where there are no direct interactions between spinons and visons that

exchange their energy. However, their semionic mutual statistics implies that the spatial arrangement of the visons affects the quantum kinetic energy of the spinons, which in turn mediates an effective, nonlocal interaction amongst the visons. We find that this interplay leads to the localisation of spinons on patches of the system—similar to quantum wells—from which the visons have been expelled in a manner comparable to the Nagaoka effect[24,25] (see also ref. [26]).

We provide an effective analytical modelling of these patches that traces their origin to a balancing act between vison configurational entropy and spinon kinetic energy. A remarkable consequence of this behaviour is that the self-localisation of spinons leads to a nonthermal, cooling-rate-dependent density of spinons. This quasiparticle excess likely manifests itself in the spin susceptibility and transport properties of the system as it is cooled from high temperatures. Since this behaviour is inherently related to both the fractionalised nature and the nontrivial mutual statistics of the excitations in the system, it is therefore an important precursor of the QSL behaviour expected at lower temperatures.

## Results

**Model.** We consider for concreteness a toric-code-inspired toy model of a gapped $\mathbb{Z}_2$ QSL. A possible microscopic derivation of the model is discussed in the Supplementary Note 1, whereas we present here only the essential features of the model in the temperature regime of interest. It can be summarised as a tight-binding model of bosonic spinons with energy cost $\Delta_s$ and hopping amplitude $t_s$ on the sites of a square lattice[27]. The visons live on the plaquettes of the lattice, with energy cost $\Delta_v$ and occupation numbers $n_p = 0$ or 1. Since the spinons and visons are mutual semions, the latter act as sources of flux of magnitude $\pi$, i.e., $\Phi_p = \pi n_p$,

$$H_s(\{n_p\}) = -t_s \sum_{\langle ij \rangle} e^{iA_{ij}} b_i^\dagger b_j + \Delta_s \sum_i b_i^\dagger b_i, \qquad (2)$$

where $b_i$, $b_i^\dagger$ obey the usual hardcore bosonic statistics, $A_{ij} = -A_{ji}$, and $(\nabla \times A)_p = \Phi_p$. Within the Born–Oppenheimer approximation, the spinons remain in their instantaneous eigenstates, with energy $E_s(\{n_p\})$, as different visons configurations $\{n_p\}$ are sampled stochastically, therefore providing an effective energy for the latter. Both spinons and visons are created or annihilated in pairs by virtue of their fractionalised nature.

Generally, one expects spinons in a random $\pi$-flux background to be weakly localised (for a recent study, see ref. [28]). At the temperatures considered in this manuscript, the spinons are sparse and the hardcore constraint makes it reasonable on energetic grounds that they will be localised far away from one another. It is therefore sensible in the first instance to investigate the problem of a single isolated spinon. We will later discuss how the results may be extended to the thermodynamic limit with a finite density of spinons.

In order to gain insight into the behaviour of the system, we perform parity-conserving Monte Carlo (MC) simulations of the stochastic ensemble of visons, $\{n_p\}$, on a square lattice containing $L \times L$ sites with periodic boundary conditions, combined with exact diagonalisation of the spinon tight-binding Hamiltonian $H_s(\{n_p\})$ (further details are given in Methods).

**Localisation of spinons.** The behaviour of the system is most intuitively illustrated by a snapshot of the vison configuration and of the corresponding spinon ground state probability density in thermodynamic equilibrium at temperature $T$, as shown in Fig. 1. The spinons are clearly localised in circular patches from which the visons have been totally expelled.

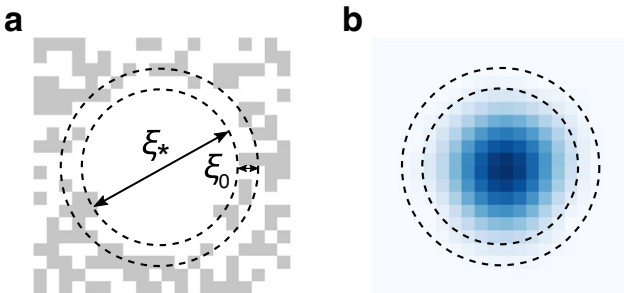

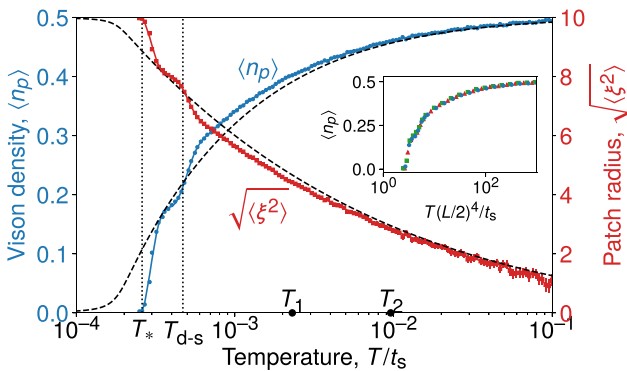

**Fig. 1 Equilibrium vison configuration and corresponding spinon ground state density. a** Vison configuration, $\{n_p\}$. The visons form an empty circular patch surrounded by a disordered background. The inner dashed line corresponds to the saddle point radius $\xi_*$ of the effective free energy (3), while the outer dashed line equals the characteristic extent of the spinon wave function, $\xi_* + \xi_0$. **b** Ground state spinon density. The data are taken from the MC simulations at $T/t_s = 10^{-3}$ for a system of size $L^2 = 20^2$ with periodic boundary conditions.

**Fig. 2 Evolution of equilibrium vison density.** We show the average vison density per plaquette, $\langle n_p \rangle$ (blue circles, left vertical scale), and the typical vison-depleted patch radius $\sqrt{\langle \xi^2 \rangle}$ (red squares, right vertical scale) as a function of temperature. The numerical data are compared with the predictions (dashed lines) of the circular disc free energy given in Eq. (3). As temperature is lowered, the finite system makes a transition to a system-spanning strip state (see Supplementary Note 2) at a temperature $T_{d\text{-}s}$, and becomes vison-free below $T_*$. The solid lines through the MC data are a guide to the eye. The calculations were performed on a system of size $L^2 = 20^2$ satisfying periodic boundary conditions. The inset shows a scaling collapse of $\langle n_p \rangle$ as a function of $T(L/2)^4/t_s$ for $L = 16$ (green squares), 18 (red triangles), 20 (blue circles).

We can understand this phenomenon in terms of a competition between the spinon kinetic energy, which favours regions with a low vison density[28], and the vison mixing entropy, which favours a uniform vison density. At finite temperature the balance produces regions of the system from which the visons are expelled, thus providing most of the support for the spinon wave function. In the complementary region, the spinon wave function is exponentially suppressed[29–32] and the visons are in a trivial, noninteracting state.

To confirm this intuition, we propose a toy one-spinon model consisting of an empty circular patch of radius $\xi$, to which the spinon is confined, while the exterior of the disc is thermally populated with visons, i.e., $p(n_p) \propto e^{-n_p \beta \Delta_v}$. The characteristic free energy $F(\xi)$ of the system as a whole is then given by

$$F(\xi) = \frac{j_0^2 t_s}{(\xi + \xi_0)^2} + \pi T \xi^2 \ln\left(1 + e^{-\beta \Delta_v}\right). \tag{3}$$

The first term describes the kinetic energy of the spinon, while the latter corresponds to the entropy of the exterior vison configurations (henceforth, we send $\Delta_v \rightarrow 0$ as it is negligible at the temperatures of interest). The prefactor $j_0^2 t_s$ is set by the ground state energy of an infinite circular well, where $j_0$ denotes the first zero of the Bessel function $J_0(x)$. The energy gap between the ground state and the first excited state of the quantum well is much larger than the temperature of interest, and thus we assume the spinon to be in the ground state. The phenomenological parameter $\xi_0$ represents effectively the penetration depth of the spinon wave function into the vison-rich region. We extract $\xi_0$ numerically by plotting the energy $E(\xi) = j_0^2 t_s (\xi + \xi_0)^{-2}$ as a function of disc radius $\xi$, averaged over exterior vison configurations (see Methods section). There are then no adjustable parameters left in the model.

Minimising (3) with respect to $\xi$ yields the typical disc radius $\xi_* \sim T^{-1/4}$ when $\xi_* \gg \xi_0$. To capture thermal fluctuations in the radius $\xi$, we estimate $\xi_*$ using,

$$\xi_*^2 \equiv \langle \xi^2 \rangle = \frac{1}{Z} \int_0^R d\xi\, \xi^2 e^{-\beta F(\xi)}, \quad Z = \int_0^R d\xi\, e^{-\beta F(\xi)}, \tag{4}$$

where $R$ is a cut-off that captures the effect of finite system size in the MC simulations. Other observables may be computed in the same vein.

In Fig. 2 we show the MC data for the average vison density $\langle n_p \rangle$ and the typical patch radius $\xi_*$ for a system of size $L^2 = 20^2$. The vison density is a monotonic function of temperature, and it

becomes vanishingly small below a characteristic temperature $T_*$: As the temperature is lowered, the spinon kinetic energy becomes dominant in the free energy and the vison-depleted patch grows. This behaviour continues until the size of the patch becomes comparable to the size of the system. We find good agreement between the MC simulation and the toy model for $T > T_*$. In our MC simulations on systems of finite size, there exists a competing vison configuration in which, rather than forming a disc, the spinon density (and the corresponding vison-depleted region) forms a strip that wraps around the torus in one direction. Such a configuration typically has a lower vison density and is responsible for the kink observed in the data at the temperature $T_{d\text{-}s}$ (see Supplementary Note 2).

The connected correlator $C_\rho(\mathbf{r}_p, \mathbf{r}_{p'}) = \langle n_p n_{p'} \rangle - \langle n_p \rangle \langle n_{p'} \rangle$ is plotted for a range of separations $\mathbf{r}_p - \mathbf{r}_{p'}$ in Fig. 3. The overall agreement between the numerical results and the toy model over a range of distances and temperatures demonstrates that our intuitive picture is indeed correct. The visons remain correlated over a characteristic distance $2\xi_*$, the typical diameter of the vison-depleted patch, which shrinks with increasing temperature [see Methods for details of the calculations using the disc model, Eq. (4)]. This picture is not modified qualitatively upon addition of weak short-ranged spinon–vison interactions (see Supplementary Note 3).

The toy model (3) predicts that $\xi_* \propto T^{-1/4}$. In a finite system of size $L^2$, this means that the vison density vanishes below a critical temperature $T_* \sim t_s L^{-4}$, as we indeed observe in the scaling collapse in the inset of Fig. 2. By contrast, a thermodynamically large system always contains a nonzero density $\rho_s$ of spinons. In this case, since the spinons are effectively hardcore bosons, we expect the visons to form a density $\rho_s$ of independent empty circular patches. This construction applies to the dilute limit where the patch size is significantly smaller than the average distance between spinons, $\xi_* \ll \rho_s^{-1/2}$. Since the thermal spinon density $\rho_s \sim e^{-\beta \Delta_s}$ vanishes exponentially fast as $T$ decreases[33], whereas $\xi_*$ increases only algebraically, the condition is expected to hold in the temperature window of interest (1).

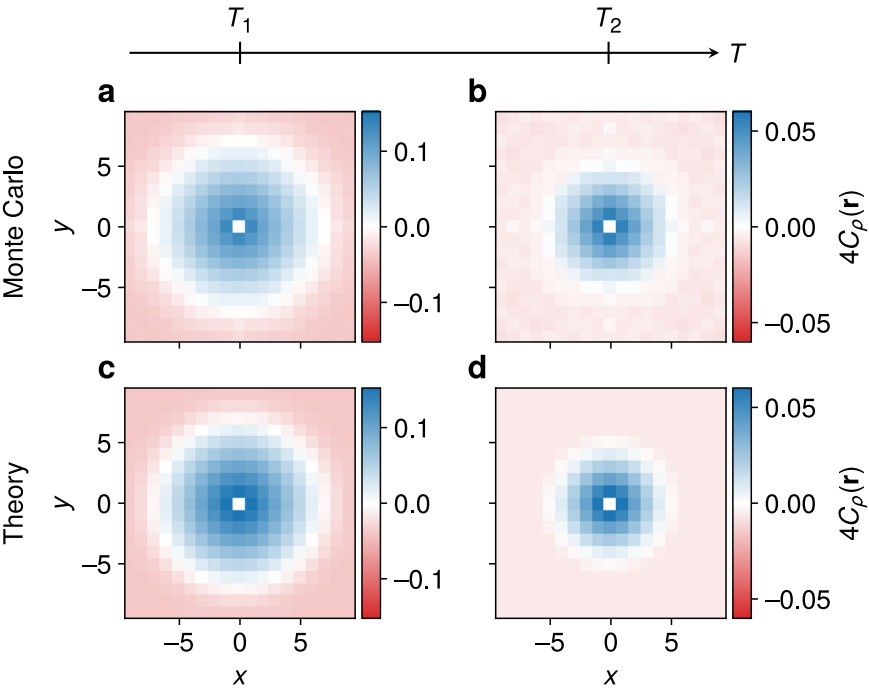

**Fig. 3 Equilibrium real-space vison correlations.** Connected vison correlator $C_\rho(\mathbf{r})$, $\mathbf{r} = (x, y)$, at two temperatures (marked for reference also in Fig. 2), **a**, **c** $T_1/t_s = 2.3\,10^{-3}$ and **b**, **d** $T_2/t_s = 9.6\,10^{-3}$. **a**, **b** correspond to the MC data, while **c**, **d** are the predictions of the empty disc model. As temperature is increased, the typical size of the vison-depleted patch shrinks and the length scale over which the visons are correlated is correspondingly reduced. The calculations were performed for a system of size $L^2 = 20^2$ satisfying periodic boundary conditions.

**Thermal quenches**. The self-localisation of spinons has a number of interesting consequences. Suppose we initialise the system in thermodynamic equilibrium at some finite temperature $T_0$, where the condition discussed above, $\xi_* < \rho_s^{-1/2}$, is satisfied. Let us then lower the temperature at a constant rate and follow the evolution of the spinon density $\rho_s$. The largest energy scale relevant to spinons is their cost $\Delta_s$, and one therefore expects $\rho_s \sim e^{-\beta\Delta_s}$ if the process is adiabatic. However, the spinons are localised in well-separated patches. To remain in equilibrium as the temperature is lowered, the spinons must annihilate with one another pairwise to reduce their density. They have two annihilation pathways: via tunnelling between two patches—a process which is suppressed in distance due to the localisation of the spinon wave function—or via motion of the patches. The latter process is also slow since it requires a coordinated change in the vison configuration without any energetic driving. Hence, if the cooling rate is sufficiently large, spinon annihilation processes cannot maintain equilibrium and $\rho_s$ develops a plateau.

On the other hand, as the temperature is lowered, the patches continue to grow at a comparatively fast rate, since the process merely requires the (energetically favourable) pairwise annihilation of visons at the edge of each patch. This will progress until the patches eventually come within reach of one another and the spinon annihilation can resume on timescales that are fast compared to the temperature variation. This happens at the threshold $T_* \sim t_s\rho_s^2$. From this time onwards, the spinon density $\rho_s$ resumes its decay; however, it is kinematically locked to the temperature via the relation $T \sim t_s\rho_s^2$. In other words, the spinon density now decreases at an anomalous, out-of-equilibrium rate, $\rho_s \sim \sqrt{T}$. A simple stochastic modelling to illustrate this out-of-equilibrium behaviour is presented in Supplementary Note 5.

Notice that, at this point, if one were to reverse the direction of the temperature variation, upon increasing $T$ the patches shrink and the spinon density $\rho_s$ again remains fixed at a value that is much higher than its equilibrium counterpart. This plateau persists until the temperature $T_{th}$ is reached, where $\rho_s \simeq e^{-\Delta_s/T_{th}}$, at which point the density resumes increasing along the adiabatic curve. One can therefore engineer corresponding hysteretic loops, illustrated schematically in Fig. 4.

We note that the plateaux in $\rho_s$ not only signal a thermodynamic quantity being invariant, but also indicate to a large extent that the positions of the spinons (vison-depleted patches) do not change (their drift motion being a slow process), leading to remarkable memory effects. Any experimental techniques that provide access to the spinon density or its spatial correlators will likely measure signatures of this hysteretic, nonequilibrium behaviour. For instance, the spinon density $\rho_s$ can be directly related to the magnetic susceptibility, $\chi \sim \rho_s$, which can be probed either by thermodynamic measurement or nuclear magnetic resonance through the Knight shift[34,35]. In thermal equilibrium, $\rho_s$ is exponentially suppressed due to the large spinon gap. However, if the system is cooled rapidly, the aforementioned nonthermal evolution of the spinon density $\rho_s$ manifests itself in an enhancement of $\chi$ with respect to the equilibrium value, which may be detected in experiments.

Our results can also be expected to have significant repercussions on transport properties where visons and/or spinons contribute (e.g., thermal transport[36]). The largest effect will likely be from the vison density (and thence their flux), which is reduced by a factor $1 - \pi\rho_s\langle\xi^2\rangle$ due to the spinon patches, and correspondingly acquires a modified temperature dependence. On the other hand, we have already discussed how the spinon motion is expected to be slow, either via tunnelling from one patch to another area of the system that happens to be sufficiently vison-depleted, or via patch drift. This behaviour is in stark contrast with the regime in which the visons are sparse or absent and spinons can propagate freely throughout the system.

## Discussion

We studied the implications of nontrivial mutual statistics and fractionalisation on excitation densities and their correlations in

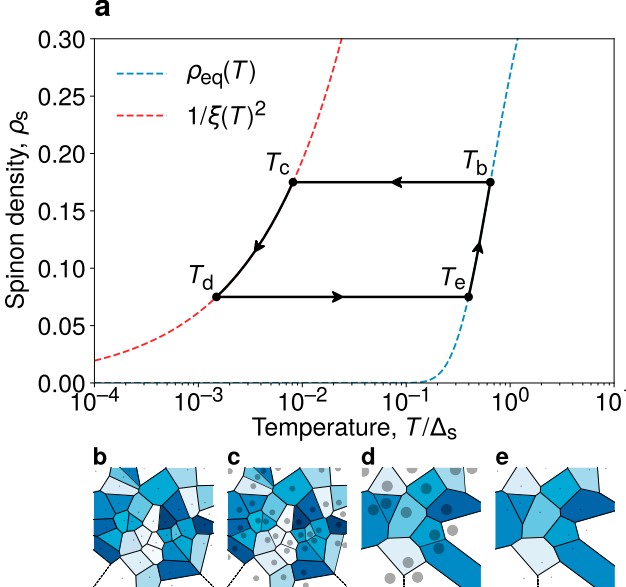

**Fig. 4 Nonequilibrium response of the spinon density to a temperature cycle. a** Schematic illustration of the nonequilibrium spinon density. The system is initially prepared in equilibrium at temperature $T_b$. If temperature is then lowered at a sufficiently large rate, $\rho_s$ falls out of equilibrium—the patches grow in diameter but their diffusive motion is slow. At $T_c$, the typical separation of the patches becomes comparable to their radius $\xi$, and the closest pairs begin to annihilate. The density then remains kinematically locked to $1/\xi^2$ as pairs continue to annihilate. If the direction of temperature variation is then reversed at $T_d$, $\rho_s$ develops another plateau as pairwise annihilation of spinons ceases and the diameter of the patches shrinks. This continues until a sufficiently high temperature, $T_e$, is reached at which thermodynamic equilibrium is restored. The behaviour of the vison-depleted patches at each of these temperatures is depicted in **b**–**e**. The patches are qualitatively represented by the solid circles and identified by the colour of their Voronoi cell.

toric-code-inspired $\mathbb{Z}_2$ QSLs at finite temperature. We considered a temperature regime of particular experimental interest in which the low-energy visons are populated thermally, while the energetically costly spinons hop coherently. The balance of spinon kinetic energy and vison configurational entropy leads to the emergence of vison-depleted patches in which the spinons remain localised. Similarly to the way in which ferromagnetic order is favoured by the kinetic energy of a single hole in the Nagaoka effect, here the kinetic energy of a spinon favours vison-free regions in the system. The size of the patches is determined by temperature, with a typical radius that scales as $T^{-1/4}$.

We highlighted important consequences of this phenomenon in the nonequilibrium behaviour of the system in response to temperature ramps. The diffusive motion of the patches is slow, whilst the rate at which they can grow or shrink is energy-driven and hence significantly faster. Since the spinons must annihilate pairwise, this means that the spinon density $\rho_s$ readily falls out of equilibrium upon cooling the system and becomes kinematically locked to $\rho_s \sim \sqrt{T}$. The excess of spinons with respect to their equilibrium density at the same temperature directly affects experimentally relevant quantities such as the magnetic susceptibility and transport properties. Since the effect is inherently due to the combination of nontrivial mutual statistics and fractionalisation of the excitations, its observation would represent an important fingerprint of QSL behaviour. Furthermore, the localisation of spinons on mobile patches would also serve as an indirect signature for the visons, which have hitherto remained elusive in experiments[37].

While the effective model that we discuss, Eq. (2), is derived using perturbation theory (see Supplementary Note 1), we expect that our main conclusions will be applicable outside of this pertubative limit. Indeed, the phenomena that we have described are a direct consequence of (i) the mutual statistics between spinons and visons, and (ii) the separation of energy scales. There is hence a strong reason to believe that these phenomena are robust even when the quantum fluctuations are more appreciable, so long as those prerequisites hold. In particular, spinons and visons remain good quasiparticles as long as the system is not in the immediate vicinity of a confinement/Higgs transition.

So far we have ignored for simplicity any interaction terms between the quasiparticles. While these terms are generally expected, so long as they do not cause the quasiparticles to condense, they only affect the phenomena we discuss quantitatively and not qualitatively. Indeed, interactions between visons would merely alter the form of the classical entropic term in Eq. (3); and short-ranged interactions between spinons are altogether negligible in the regime where the size of their patches exceeds the characteristic interaction length scale. The only couplings worth investigating in detail are those between spinons and visons, through which the latter can act as diagonal disorder for the former thence also leading to localisation. As we discuss in Supplementary Note 3, in systems satisfying the condition (1), this effect alone is too weak to lead to the formation of well-defined depleted patches.

It is interesting to draw an analogy between the mechanism discussed in our work and the behaviour of type-I superconductors. Indeed, the expulsion of visons from spinon patches operates in a similar manner to the Meissner effect where magnetic vortices are expelled from the superconductor, driven in both cases by a reduction in the quantum kinetic energy of the system[38]. The fact that a very closely related mechanism operates robustly in real materials, leading to experimentally measurable properties, supports the claim that our results are not inherently limited to the theoretical model considered in our work.

We therefore expect our results to apply to gapped $\mathbb{Z}_2$ spin liquid candidate materials. For the gapless $\mathbb{Z}_2$ spin liquids hosted by Kitaev materials, there exists a temperature regime similar to Eq. (1), where the spinons remain quantum coherent whereas the visons are thermally populated. It would be interesting to examine to what extent our results may be generalised to this case.

We note that interference effects also play a role in topological systems with more exotic statistics between the quasiparticles. As shown in ref. [28], one may generally expect localisation effects, although there are important quantitative differences with respect to the time-reversal-symmetric $\mathbb{Z}_2$ case. Moreover, if we consider for instance $\mathbb{Z}_N$ theories, the entropy of the exterior vison configuration in Eq. (3) increases, $S \propto \ln N$, favouring a smaller vison-depleted region. All these, as well as the case of non-Abelian statistics, are interesting directions for future work.

Other interesting and open questions include the role of disorder, in particular on transport properties, if it is capable of localising the spinons or pinning the visons in a way that significantly alters the circular shape of the patches. One could also consider how the mechanism generalises to higher-dimensional systems ($d > 2$), both in topological as well as fractonic systems. Mutual statistics is likely to produce similar interference effects; however, dimensionality will play an important role, in particular because topological quasiparticles embedded in higher dimensions usually take the form of extended objects (e.g., closed loops or membranes). Understanding how these quasiparticles may become localised is a challenging and interesting question in its own right[39].

Finally, in our simulations we observed an instability in the shape of the patches at low temperature (from circular to strip-like, see Supplementary Note 2). While in our case it is merely a finite size effect due to the spinon wave function overlapping with itself across the periodic boundary conditions, it nonetheless suggests that a similar (possibly nematic) instability may occur in a thermodynamic system when the patches approach one another. Investigating this instability is an interesting future direction, as it affects the spectral properties of the spinons, and possibly alters in a measurable way the response properties of the system.

## Methods

**Monte Carlo simulations.** The thermal average of an observable $\mathcal{O}$ that is diagonal in the plaquette operators assumes the form

$$\langle \mathcal{O} \rangle = \frac{1}{Z} \sum_{\{n_p\}} \mathrm{Tr} \mathcal{O}(\{n_p\}) \exp[-\beta H_s(\{n_p\}) - \beta N_v \Delta_v], \quad (5)$$

where $H_s(\{n_p\})$ is the spinon tight-binding Hamiltonian (2), the trace is over the spinon degrees of freedom given a vison configuration $\{n_p\}$, and $N_v = \sum_p n_p$ is the total vison number. $Z = \sum_{\{n_p\}} e^{-\beta N_v \Delta_v} \mathrm{Tr} e^{-\beta H_s(\{n_p\})}$ is the partition function of the system.

Averages of the form (5) can be evaluated efficiently using Markov chain Monte Carlo (MC) applied to the vison degrees of freedom $\{n_p\}$. The proposed updates of the system must however respect the constraint (when imposing periodic boundary conditions) that the total flux threading the lattice, $\Phi_t = \sum_p \pi n_p$, equals an integer multiple of $2\pi$ (equivalently, the total number of vison excitations must be even). Note that the global fluxes threading the torus are chosen to vanish. We make use of the following discrete update, which explicitly preserves the parity of the total number of vison excitations:

(i)   choose two plaquettes $p$, $p'$ (with $p \neq p'$) at random, and propose the corresponding update to the vison configuration:

$$n_p \to n_p' \equiv 1 - n_p;$$
$$n_{p'} \to n_{p'}' \equiv 1 - n_{p'};$$

(i)   construct the new spinon tight-binding Hamiltonian $H' \equiv H_s(\{n_p'\})$ by drawing a string $\gamma_{pp'}$ between the two flipped plaquettes, i.e., setting $A_{ss'} \to A_{ss'} + \pi$ along the bonds belonging to the path, $\langle ss' \rangle \in \gamma_{pp'}$;

(ii)  diagonalise the new spinon Hamiltonian $H_s'$ to obtain the full energy spectrum;

(iii) accept the proposed update according to the Metropolis acceptance probability: $\min(1, \ \mathrm{tr} e^{-\beta H'}/\mathrm{tr} e^{-\beta H})$, where $H = H_s + N_v \Delta_v$.

The initial state of the system is set using a random distribution of visons living on the plaquettes with density $\rho_v = 1/2$ (using even system sizes only, which implies that $\rho_v L^2$ is even, as required). The system is then gradually cooled using $O(10^4)$ MC sweeps, where one MC sweep of the system is equal to $L^2/2$ individual MC steps of the form (i)–(iv). For example, in our simulations of a system of size $L = 20$, decreasing temperatures $T_n$ are taken between $T/t_s = 0.1$ and $T/t_s = 2.5 \ 10^{-4}$, in $2^7$ logarithmically-spaced increments, with an equilibration time $t_n = \lceil 4 \exp(\alpha/T_n) \rceil$, where $\alpha$ is chosen such that $\sum_n t_n \sim 10^4$. Measurements are then made after this time at each temperature $T_n$. The parameters in the above cooling protocol are chosen to ensure that the system remains in equilibrium for each measurement. This was checked by calculating the system's characteristic relaxation time, deduced from the decay of the vison autocorrelation function, at several temperatures throughout the cooling protocol (taking care to account for metastability). Finally, the results are averaged over $2^9$ independent cooling histories.

In the limit $\beta \Delta_v \ll 1$, the vison energy cost can be safely neglected. We have indeed checked explicitly that adding a small vison chemical potential contribution to the energy of the system does not alter our results quantitatively. Further, one may show using the effective disc free energy that our results are likely to be qualitatively unchanged as long as $\Delta_v \lesssim T_* \sim t_s/L^4$. The vison chemical potential only has an appreciable effect when $T \gtrsim \Delta_v \gtrsim T_*$, in which case, the energetic (rather than entropic) cost of visons becomes substantial, and consequently their density is trivially suppressed.

**Spinon ground state energy.** The empty disc model assumes that the spinon energy $E(\xi)$, corresponding to a disc of radius $\xi$ surrounded by disordered visons, can be parametrised as

$$E(\xi) = \frac{j_0^2 t_s}{(\xi + \xi_0)^2}. \quad (6)$$

The numerator $j_0^2 t_s$ is fixed by the large $\xi$ behaviour—in this limit, the energy should be asymptotically described by that of a free particle in an infinite circular well of radius $\xi$. Hence, $j_0$ is the first zero of the Bessel function $J_0(x)$. The

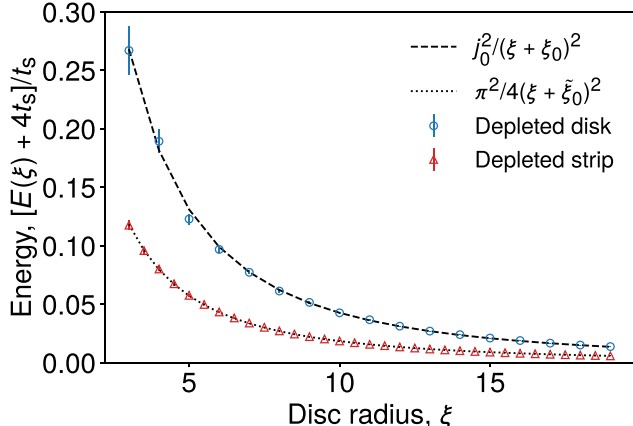

**Fig. 5 Parameterising the spinon ground state energy.** Average ground state energy of a spinon subjected to a vison distribution in which there exists an empty disc of radius $\xi$ (blue circles), or the vison-depleted region forms a strip of width $2\xi$ that wraps around the torus in one direction (red triangles), surrounded by a disordered region of $\pi$-fluxes with average density 1/2. The dashed and dotted lines correspond to the best fit to functions shown in the legend. The data are averaged over 250 flux realisations in a system of size $L^2 = 40^2$, and the error bars denote the standard deviation of the energy at a given disc radius, not the error in the mean. The parametrisation chosen for $E(\xi)$ on the vertical axis is merely a matter of convenience.

parameter $\xi_0$ represents phenomenologically the penetration depth of the spinon wave function into the disordered vison background surrounding the empty circular patch.

In order to fix the value of $\xi_0$, we sample random configurations of visons in which there exists an empty disc of radius $\xi$, and in the complementary region the visons appear randomly with probability 1/2 per plaquette:

$$p(n_p) = \begin{cases} 0 & \text{if } |\mathbf{r}_p| < \xi, \\ \frac{1}{2} & \text{otherwise}. \end{cases} \quad (7)$$

The resulting ground state energy of the spinon is then averaged over the exterior vison configurations. The resulting averaged energy is plotted in Fig. 5, and a fit to Eq. (6) is performed, leading to the value $\xi_0 = 1.64(2)$. This value does not exhibit significant variation with system size $L$.

The same method may be applied to the strip vison configuration (discussed further in Supplementary Note 2), also shown in Fig. 5, in which the spinon wave function wraps around the torus in one direction. There are two such configurations in a square system with periodic boundary conditions. The energy of a strip of width $2\xi$ may be parametrised as

$$E(\xi) = \frac{\pi^2 t_s}{4(\xi + \tilde{\xi}_0)^2}. \quad (8)$$

Fitting the numerical data with this function gives a value $\tilde{\xi}_0 = 1.565(3)$.

**Effective empty disc model.** The average area $\pi \langle \xi^2 \rangle$ of the vison-depleted patch at a given temperature $T = \beta^{-1}$ may be calculated using the disk free energy in Eq. (3):

$$\langle \xi^2 \rangle = \frac{1}{Z} \int_0^R d\xi \ \xi^2 e^{-\beta F(\xi)}, \quad (9)$$

where $Z = \int_0^R d\xi \ e^{-\beta F(\xi)}$. Since for temperatures satisfying $T \gg \Delta_v$ the exterior region has a vison density of 1/2, the average vison density over the system as a whole is

$$\langle n_p \rangle = \frac{1}{2}\left(1 - \frac{\langle \xi^2 \rangle}{R^2}\right). \quad (10)$$

Further, since the model assumes that the vison occupation numbers are perfectly correlated within the empty patch, and uncorrelated outside, we may approximate the connected vison correlator in the following way. For two plaquettes separated by the vector $\mathbf{r}$, with $r = |\mathbf{r}|$, the number of correlated pairs that reside within the disc of radius $\xi$ is given by

$$A(r; \xi) = \Theta(2\xi - r)\left[2\xi^2 \arccos\left(\frac{r}{2\xi}\right) - \frac{r}{2}\sqrt{4\xi^2 - r^2}\right], \quad (11)$$

i.e., the area of intersection of two circles, each with radius $\xi$, whose centres are

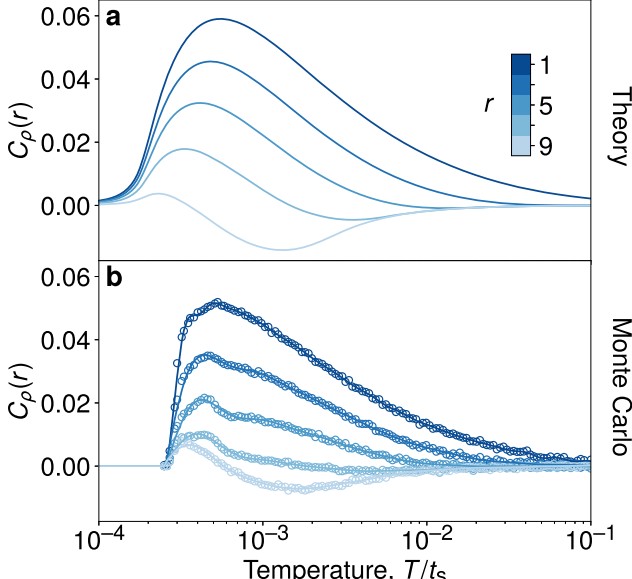

**Fig. 6 Temperature dependence of real-space vison correlations.** We show the vison density correlator $C_\rho(r)$ as a function of temperature at distances $r = 1, 3, ..., 9$ for a system of size $L^2 = 20^2$. **a** Theoretical predication from Eq. (12) using the disc free energy in Eq. (3). **b** The corresponding MC data. The solid lines through the MC data are a guide to the eye.

separated by a distance $r$. $\Theta(x)$ is the Heaviside step function. As required, $A(r; \xi)$ vanishes for $r > 2\xi$, and $A(0; \xi) = \pi\xi^2$. The density-density correlator may then be approximated by the cylindrically symmetric function

$$C_\rho(r) \simeq \frac{\langle A(r; \xi)\rangle}{A(r; R)} - \frac{\langle\xi^2\rangle^2}{R^4}. \qquad (12)$$

The predictions of Eqs. (10) and (12) are plotted in Figs. 2 and 3, respectively. In Fig. 6 we compare the analytical expression for the correlator $C_\rho(r)$ as a function of temperature, for a range of distances $r$, with the corresponding MC data.

## Data availability
The data that support the findings of this study are available from the corresponding author upon reasonable request.

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

## Acknowledgements
The authors are grateful to Claudio Chamon for several interactions and in particular for suggesting the form of the effective spinon free energy. We would also like to thank Fabian Essler, Austen Lamacraft, Max McGinley, Roderich Moessner and Oleg Tchernyshyov for useful discussions. This work was supported in part by Engineering and Physical Sciences Research Council (EPSRC) Grants No. EP/P034616/1 and No. EP/M007065/1 (C.C. and O.H.), National Natural Science Foundation of China, Grant No. 11974396 and Strategic Priority Research Program of the Chinese Academy of Sciences, Grant No. XDB33020300 (Y.W.). The numerics were performed using resources provided by the Cambridge Service for Data Driven Discovery (CSD3) operated by the University of Cambridge Research Computing Service (http://www.csd3.cam.ac.uk/),

provided by Dell EMC and Intel using Tier-2 funding from the Engineering and Physical Sciences Research Council (capital grant EP/P020259/1), and DiRAC funding from the Science and Technology Facilities Council (https://dirac.ac.uk/).

## Author contributions
All authors (O.H., Y.W. and C.C.) contributed to the formulation of the study, interpretation of the results and writing of the manuscript. O.H. developed and performed the calculations and numerical simulations.

## Competing interests
The authors declare no competing interests.
