## [Peer Review File · Nature Communications]

Reviewer 1 comments:

In my opinion, while the referee has raised some valid concerns, the authors have satisfactorily addressed those concerns and the manuscript should be published in its current version.

Specifically, points 2, 3, and 4 raised by the referee have been clearly addressed by the authors. Whether point 1 has been satisfactorily addressed is a more subjective matter. In essence, the referee points out that a particular assumption made by the authors - the separation of energy scales - is not expected to occur generically in physical materials. Here, I agree with the referee that there is indeed no intrinsic reason to expect that such a separation of scales occurs generically. However, as pointed out by the authors, if a certain material hosts a stable gapped QSL ground state, then one expects a parametric separation of scales on fundamental grounds and the results are expected to hold. Moreover, it is likely that gapped QSL ground states will be realized in noisy intermediate-scale quantum (NISQ) devices which can implement e.g., the toric code Hamiltonian, which is exactly solvable and thus stable against arbitrary local perturbations.

Whether or not the authors' results apply to a specific realization or a specific material candidate will only be determined after it is experimentally established whether that material harbors the parametric separation of scales required. The question at this point is then whether one plausibly expects this to indeed be the case; on this, I am in agreement with the authors and think they have convincingly argued their case in the response.